



# Tracing and visualisation of contributing water sources in the LISFLOOD-FP model of flood inundation

Matthew D. Wilson[1,2] and Thomas J. Coulthard[3]

[1]Geospatial Research Institute | Toi Hangarau, University of Canterbury, New Zealand
[2]School of Earth and Environment | Te Kura Aronukurangi, University of Canterbury, New Zealand
[3]Department of Geography, Geology and Environment, University of Hull, United Kingdom

**Correspondence:** Matthew Wilson (matthew.wilson@canterbury.ac.nz)

**Abstract.** We describe the formulation of a simple method of water source tracing for computational models of flood inundation and demonstrate its implementation within CAESAR-Lisflood. Water source tracing can provide additional insight into flood dynamics by accounting for flow pathways. The method developed is independent of the hydraulic formulation used, allowing it to be implemented in other model codes without affecting flow routing. In addition, we developed a method which allows

up to three water sources to be visualised in RGB colour-space, while continuing to allow depth to be resolved. We show the application of the methods developed for example applications of a major flood event, a shallow estuary, and Amazonian wetland inundation. A key advantage of the formulation developed is that the number of water sources which may be traced is limited only by computational considerations. In addition, the method is independent of the hydraulic formulation, meaning that it is relatively straightforward to add to existing finite volume codes including those based on or developed around the

LISFLOOD-FP method.

## 1 Introduction

Flood inundation models relate upstream river inflow and other dynamic boundary conditions (e.g. downstream stage) to inundation extent, depth and velocity, and have become invaluable tools for the assessment and understanding of flood dynamics and risk. Recently a series of fast and effective two dimensional hydrodynamic models have been developed (Hunter et al.,

2007; Neal et al., 2018). These model codes have enabled assessment of flood inundation at high spatial resolutions (e.g. Yu and Coulthard, 2015) and large spatial scales including applications on the Amazon (Wilson et al., 2007) and Congo (O'Loughlin et al., 2020), at continental scales (Dottori et al., 2021; Wing et al., 2017) and global (Dottori et al., 2016; Sampson et al., 2015). The LISFLOOD-FP model (Bates and De Roo, 2000; Bates et al., 2010) uses a fixed raster grid structure and includes several flow formulations of different physical complexity (Shaw et al., 2021), which have been widely used for flood hazard

assessment in fluvial (Horritt et al., 2010) and coastal (Vousdoukas et al., 2018) applications, as well as within models of landscape evolution (Adams et al., 2017; Coulthard et al., 2013). However, the ability to trace the sources of water in the model domain is presently missing from reduced-complexity 2D flood models such as LISFLOOD-FP.

Understanding the contribution of different water sources to flooding and river flows is important when managing river basins, for example determining the relative contribution of tributaries or where water borne contamination is an issue. This





is a function found in more complex two- and three-dimensional models based on the full shallow water equations, such as TELEMAC (Galland et al., 1991) and Ansys Fluent, and has been used for a variety of applications including water quality modeling in a lake (Kopmann and Markofsky, 2000) and solute and viral dispersal in estuaries (Robins et al., 2014, 2019). For example in TELEMAC, non-buoyant tracers can be added and their course and concentration followed (Ch. 9, Ata et al., 2014). However, such 2 and 3D codes have a considerable computational overhead compared to simpler schemes such as

LISFLOOD-FP.

In this paper, we build on the work of Wilson and Coulthard (2019) to propose a simple and efficient method which can be added to finite volume codes to track the contribution of multiple water sources to predicted flood depths, and demonstrate it within the CAESAR-Lisflood model (Coulthard et al., 2013) for several example flooding case studies which each represent the mixing of water from multiple sources. CAESAR-Lisflood[1] is a development of the CAESAR model (Coulthard et al.,

2002; Van De Wiel et al., 2007) and the LISFLOOD-FP 2D hydrodynamic model (Bates et al., 2010) that simulates landscape evolution by coupling a hydrological model, a surface water flow model, fluvial erosion and deposition and slope processes. Here, we focus only on surface water, and additionally develop a method for the visualisation of water sources.

## 2 Methods

### 2.1 Flow formulation

CAESAR-Lisflood implements the LISFLOOD-FP inertial formulation (Bates et al., 2010) to estimate depths across the domain. This is a reduced physical complexity mass-conservative hydraulic model structured on a Cartesian grid, in which the updated water depth $h$, at time $t$, for cell $i,j$ is determined using (Bates and De Roo, 2000):

$$h_{i,j}^t = h_{i,j}^{t-\Delta t} + \Delta t \frac{Q_{x,i-1,j}^t - Q_{x,i,j}^t + Q_{y,i,j-1}^t - Q_{y,i,j}^t}{\Delta x^2} \tag{1}$$

where $h_{i,j}^{t-\Delta t}$ is the cell depth at the end of the previous timestep ($\Delta t$), $Q$ represents the flows into or out of the cell in the

$x$ or $y$ directions, and $\Delta x$ is the cell size. Flows are decoupled in each direction; flow in the $x$ direction is determined using (de Almeida et al., 2012; Bates et al., 2010):

$$Q_x^t = \frac{q_x^{t-\Delta t} - gh_{[\text{flow}]}^t \Delta t \frac{\Delta(h^t+z)}{\Delta x}}{(1 + gh_{[\text{flow}]}^t \Delta t n^2 |q_x^{t-\Delta t}|/(h_{[\text{flow}]}^t)^{10/3})} \Delta y \tag{2}$$

with flow in the $y$ direction obtained analogously. In Eqn. (2), $q_x^{t-\Delta t}$ represents the flux between the cells from the previous timestep ($Q_x^{t-\Delta t}/\Delta y$), $g$ is acceleration from gravity, $h_{[\text{flow}]}$ is the maximum depth of flow between two cells, $n$ is Manning's

roughness and $z$ is the cell bed elevation. The LISFLOOD-FP inertial formulation has been benchmarked against other formulations and industry standard codes and showed that, in appropriate flow conditions (i.e. gradually varied flow, Froude number $< \sim 1$), the model performed favourably and with high efficiency (de Almeida and Bates, 2013; Neal et al., 2012).

---

[1]Available from https://sourceforge.net/projects/caesar-lisflood/





## 2.2 Water source tracing

Given that the sum of depths in a cell from each source, $w$, make up its total flood depth, $h$:

$$\sum_{w=1}^{W} h_w = h, \tag{3}$$

the volume fraction of each source, $\phi_w$, may be obtained from:

$$\phi_w = \frac{h_w}{h}, \tag{4}$$

ensuring:

$$\sum_{w=1}^{W} \phi_w = 1. \tag{5}$$

Thus, the fraction of each water source in each cell is defined as the depth of water from that source in the cell, divided by the total cell depth.

Following this, once flows between cells are calculated and depths updated, water tracing proceeds as follows: (1) for each cell, the remaining depth following removal of water from any outflows is found; (2) the amount of the depth belonging to each water source is obtained by scaling the depths according to water source fractions from the previous timestep; (3) contributing inflow depths from each source for all neighbours is added to each source depth; and (4) updated water source fractions are calculated as per Eqn. 4 by dividing the fractions from each source by the updated total cell depth. More formally, for each cell, for each water source, $w$, at time, $t$, the volume fraction $\phi$ is:

$$\phi_w^t = \frac{h_{Q_{[\text{out}]}}^t \phi_w^{t-\Delta t} + \sum_D (Q_{[\text{in}]}^{t,D} \phi_w^{t-\Delta t,D}) \frac{\Delta t}{\Delta x^2}}{h^t} \tag{6}$$

where $h_{Q_{[\text{out}]}}^t$ represents the depth remaining in the cell after the removal of outflows at the current time, $Q_{[\text{out}]}^t$ (and before the addition of any inflows), which is scaled according to the fraction from this source from the previous timestep, $\phi_w^{t-\Delta t}$. To this depth is added the total amount of water from this source which is contributed from neighbouring cells, given by $\sum_D (Q_{[\text{in}]}^{t,D} \phi_w^{t-\Delta t,D})$, where $Q_{[\text{in}]}^{t,D}$ is the inflow from a particular direction, $D$, and $\phi_w^{t-\Delta t,D}$ is the fraction of flow for source $w$ from that direction. The updated source volume fraction is obtained by dividing by the updated cell depth, $h^t$. Finally, $h_{Q_{[\text{out}]}}^t$ is given by:

$$h_{Q_{[\text{out}]}}^t = h^{t-\Delta t} - \sum_D Q_{[\text{out}]}^{t,D} \frac{\Delta t}{\Delta x^2} \tag{7}$$





where $Q_{[\text{out}]}^{t,D}$ are the outflows in each direction. To improve computational efficiency, the method only requires knowledge of the source volume fractions for the previous timestep in each neighbouring, upstream cell; the actual depths or volumes from each source in each cell are not saved. The calculation cell depth after removal of outflows, $h_{Q_{[\text{out}]}}^t$, enables each source fraction to be updated based on the flows in $(Q_{[\text{in}]}^{t,D})$, the fraction from each source in direction, $D$ ($\phi_w^{t-\Delta t,D}$), and the updated depth, $h^t$.

Using this approach, calculation of water source fractions flowing out of the cell is not necessary. For completeness, using the notation of (1) and expanding $D$ into cell indices, $i,j$, the updated depth from each source can be obtained using:

$$\phi_{w,i,j}^t h_{i,j}^t = \phi_{w,i,j}^{t-\Delta t} h_{i,j}^{t-\Delta t} + \Delta t \frac{\phi_{w,i-1,j}^{t-\Delta t} Q_{x,i-1,j}^t - \phi_{w,i+1,j}^{t-\Delta t} Q_{x,i,j}^t + \phi_{w,i,j-1}^{t-\Delta t} Q_{y,i,j-1}^t - \phi_{w,i,j+1}^{t-\Delta t} Q_{y,i,j}^t}{\Delta x^2}. \tag{8}$$

Boundary condition inputs are added prior to the routing of surface water. Adjusting the water volume fractions for the cells in which water is added is straightforward. The boundary modified water volume fractions, $\phi_w'$, are obtained using:

$$\phi_w' = \frac{h\phi_w + Q_w \frac{\Delta t}{\Delta x^2}}{h + \sum_w Q_w \frac{\Delta t}{\Delta x^2}} \tag{9}$$

for input sources, or:

$$\phi_w' = \frac{h\phi_w}{h + \sum_w Q_w \frac{\Delta t}{\Delta x^2}} \tag{10}$$

for other sources, where $Q_w$ is the inflow added to the cell from source $w$ at the start of timestep. Thus, fractions from sources where water is added to the cell are adjusted upwards, while fractions for non-source volumes are adjusted downwards.

It should be noted that, for simplicity, this method treats cell water volumes as fully mixed. Consequently, it may be possible for small fractions to propagate quickly in a downstream direction, since fluxes into a cell would be included in the fractions assigned as inflow to a downstream neighbouring cell in the next timestep, via $\phi_w^{t-\Delta t,D}$. As a result, caution should be given to the interpretation of very small water source fractions.

### 2.3 Water tracing implementation

Importantly, within the model code, the additional equations required for the water source tracing presented in Section 2.2 can be piggybacked on top of existing finite volume model code, requiring minimal modification of the original schemes. The water source tracing method was implemented in C#, within CAESAR-Lisflood version 1.8f (Coulthard, 2015). However, the method is readily transferable to other programming languages where different versions of LISFLOOD-FP (Bates et al., 2010) are implemented, or to other similar flow models. The additional C# code added to CAESAR-Lisflood is included in

Appendix A. Here, we summarise the algorithms developed.

Our method allows water from different sources to be tracked according to point sources (e.g. a reach input) or from spatial areas that can be user-defined (e.g. to represent different rain zones or subcatchments). CAESAR-Lisflood allows the addition





of water as point sources, or via rainfall through a hydrological model based on a spatially distributed version of TOPMODEL (Beven and Kirkby, 1979), for implementation see Coulthard and Skinner (2016) and Coulthard and Van De Wiel (2017). The

zonal input also allows tidal sources to be traced, although note that the model is depth-averaged and does not account for the higher density of saltwater. Water tracers are saved as two 3D or stacked arrays of size $W * X * Y$ grid cells, for the current and previous timestep, which are initialised as zeros. Additionally, two arrays for $\delta h \delta t$ in the $x$ and $y$ directions are used to save the volumes of water moving between cells in units of change in depth in the timestep.

The main functional flow of CAESAR-Lisflood as related to water source tracing is shown in Figure 1. For each time

step, inflows from various sources are first added. The addition of traced water requires code to be modified such that, after the addition of water from each source, the water source fractions are updated to account for the increase in volume from a source (Eqn. 9 and 10). In CAESAR-Lisflood, inflows to the modelled domain may reflect reach inputs, rainfall inputs and tidal or stage inputs, which are added in three separate functions (`reach_water_and_sediment_input()`, `catchment_water_input_and_hydrology()` and `stage_tidal_input()`, respectively). Each of these are up-

dated in a similar way using Algorithm 1.

Following the addition of inputs from sources, the implementation in CAESAR-Lisflood copies the water depths and water tracers into arrays which represent the previous timestep, $h_{t-1}$ and $\phi_{t-1}$. For the water source tracing implementation, the tracer states are saved via the function `save_tracer_states()` (Figure A3). Next, flows are updated in the function `qroute()`, for which no modifications are required (meaning that the water source tracing does not affect flow). Depths are

then updated via `depth_update()` with no modifications except that volumes of water moving between cells are stored to the $\delta h \delta t$ in $x$ and $y$ arrays, calculated using:

$$\delta h \delta t^{t,D} = Q^{t,D} \frac{\Delta t}{\Delta x} \tag{11}$$

for a single direction (C# code in Figure A4). This is used as part of the solution to (6) and (7) to update the water source tracers, enabling a simplified form to be implemented for the tracer update:

$$\phi_w^t = \frac{h_{Q_{[\text{out}]}}^t \phi_w^{t-\Delta t} + \sum_D (\delta h \delta t_{[\text{in}]}^{t,D} \phi_w^{t-\Delta t,D})}{h^t} \tag{12}$$

where:

$$h_{Q_{[\text{out}]}}^t = h^{t-\Delta t} - \sum_D \delta h \delta t_{[\text{out}]}^{t,D}. \tag{13}$$

Water source tracers are updated in the function `update_tracer_states()`, which is the main addition to the CAESAR-Lisflood code. Algorithm 2 provides details of the procedure used. For every cell in the model which contains water ($h_{i,j}^t > 0.0$),

the depth remaining in the cell after outflows ($h_{Q_{[\text{out}]}}^t$) is first calculated by subtracting outflows from each of the four flow directions. We sum $\delta h \delta t$ from each direction (previously calculated in `depth_update()`), which represent contributions to



**Algorithm 1** Updating water source fractions after adding input sources. Inputs required are the depth in the cell for the previous iteration ($h^{t-1}$), a vector of all water sources for the cell for the previous iteration ($\phi^{t-1}$), input from this source ($Q_w$), the index of this source ($w$), the number of sources ($W$), the cell size ($\Delta x$) and the timestep ($\Delta t$). A C# implementation is shown in Figure A2.

---

**Input:** $h^{t-1}, \phi^{t-1}, Q_w, w, W, \Delta x, \Delta t$

**Output:** $\phi^t$

$\quad \delta h \delta t_w \leftarrow Q_w * \Delta t / \Delta x^2$ {get change in depth resulting from this source}

$\quad h^t \leftarrow h^{t-1} + \delta h \delta t_w$ {get new depth after adding input from this source}

$\quad$ **if** $h^t > 0.0$ **then**

$\quad\quad$ **if** $h^t = \delta h \delta t_w$ **then**

$\quad\quad\quad \phi_w^t \leftarrow 1.0$ {if depth was zero, then all water in cell is from this source}

$\quad\quad$ **else if** $\phi_w^{t-1} < 1.0$ {if cell contains multiple sources} **then**

$\quad\quad\quad h_w^t \leftarrow \phi_w^{t-1} * h^{t-1}$ {get depth after addition of this source}

$\quad\quad\quad \phi_w^t \leftarrow (h_w^t + \delta h \delta t_w)/h^t$ {update this water source fraction}

$\quad\quad\quad$ **for** $src = 1$ to $W$ **do**

$\quad\quad\quad\quad$ **if** $src \neq w$ **then**

$\quad\quad\quad\quad\quad \phi_{src}^t \leftarrow (\phi_{src}^{t-1} * h^{t-1})/h^t$ {update fraction of other water sources}

$\quad\quad\quad\quad$ **end if**

$\quad\quad\quad$ **end for**

$\quad\quad$ **end if**

$\quad$ **end if**

---

depth from the right ($x$) or up ($y$): outflows will be negative for the right and up directions and positive in the left and down directions.

The procedure then updates each source for the cell, by first calculating the sum of depths added from this source (i.e. $\sum_D (\delta h \delta t_{[in]}^{t,D} \phi_w^{t-\Delta t,D})$), again by scanning the $\delta h \delta t$ arrays in each direction. Finally, the fraction is then calculated by dividing the total new depth from the source by the depth from all sources, $h^t$.

### 2.4 Water source visualisation

Visualisations of flood model outputs are often produced for depths in the form of animated maps. Here, we derived a simple colour scaling which permits up to three water sources to be visualised in RGB colour-space. For each cell, the RGB colour index in the range 0 to 255 is obtained for the desired water source, $w$, using:

$$RGB_{i,j} = 255 \cdot \phi_{i,j,w}^{\beta} \tag{14}$$





---

**Algorithm 2** Updating water source fractions after flow routing and depth update. Inputs required are the updated depth in the cell ($h^t$), water sources for the previous iteration ($\phi^{t-1}$), depth contributions horizontally ($\delta h \delta tx$) and vertically ($\delta h \delta ty$), the number of sources ($W$), and number of cells in the grid horizontally ($X$) and vertically ($Y$). A C# implementation is shown in Figure A5.

---

**Input:** $h^t, \phi^{t-1}, \delta h \delta tx, \delta h \delta ty, W, X, Y$

**Output:** $\phi^t$

  **for all** $i, j$ cells in ranges $1 \leq i \leq X$ and $1 \leq j \leq Y$ **do**

      **if** $h_{i,j}^t > 0.0$ **then**

          $\delta h \delta t \text{sum}_{Q_{[\text{out}]}} \leftarrow 0.0$ {obtain total change in depth after flows out (source not important}

          **if** $\delta h \delta tx_{i+1,j} < 0.0$, **then** $\delta h \delta t \text{sum}_{Q_{[\text{out}]}} \leftarrow \delta h \delta t \text{sum}_{Q_{[\text{out}]}} + \delta h \delta tx_{i+1,j}$; **end if** {right, -ve = outflow}

          **if** $\delta h \delta tx_{i,j} > 0.0$, **then** $\delta h \delta t \text{sum}_{Q_{[\text{out}]}} \leftarrow \delta h \delta t \text{sum}_{Q_{[\text{out}]}} - \delta h \delta tx_{i,j}$; **end if** {left, +ve = outflow}

          **if** $\delta h \delta ty_{i,j+1} < 0.0$, **then** $\delta h \delta t \text{sum}_{Q_{[\text{out}]}} \leftarrow \delta h \delta t \text{sum}_{Q_{[\text{out}]}} + \delta h \delta ty_{i,j+1}$; **end if** {up, -ve = outflow}

          **if** $\delta h \delta ty_{i,j} > 0.0$, **then** $\delta h \delta t \text{sum}_{Q_{[\text{out}]}} \leftarrow \delta h \delta t \text{sum}_{Q_{[\text{out}]}} - \delta h \delta ty_{i,j}$; **end if** {down, +ve = outflow}

          $h_{Q_{[\text{out}]}}^t \leftarrow h^{t-1} + \delta h \delta t \text{sum}_{Q_{[\text{out}]}}$ {get depth remaining in cell after removal of outflows}

          **for** $src = 1$ to $W$ **do**

               $\delta h \delta t \text{sum}_{Q_{[\text{in}]}} \leftarrow 0.0$ {obtain total change in depth after for inflows of this source (ignoring outflows)}

               **if** $\delta h \delta tx_{i+1,j} > 0.0$, **then** $\delta h \delta t \text{sum}_{Q_{[\text{in}]}} \leftarrow \delta h \delta t \text{sum}_{Q_{[\text{in}]}} + \delta h \delta tx_{i+1,j} * \phi_{i+1,j,src}^{t-1}$; **end if** {right, +ve = inflow}

               **if** $\delta h \delta tx_{i,j} < 0.0$, **then** $\delta h \delta t \text{sum}_{Q_{[\text{in}]}} \leftarrow \delta h \delta t \text{sum}_{Q_{[\text{in}]}} - \delta h \delta tx_{i,j} * \phi_{i-1,j,src}^{t-1}$; **end if** {left, -ve = inflow}

               **if** $\delta h \delta ty_{i,j+1} > 0.0$, **then** $\delta h \delta t \text{sum}_{Q_{[\text{in}]}} \leftarrow \delta h \delta t \text{sum}_{Q_{[\text{in}]}} + \delta h \delta ty_{i,j+1} * \phi_{i,j+1,src}^{t-1}$; **end if** {up, +ve = inflow}

               **if** $\delta h \delta ty_{i,j} < 0.0$, **then** $\delta h \delta t \text{sum}_{Q_{[\text{in}]}} \leftarrow \delta h \delta t \text{sum}_{Q_{[\text{in}]}} - \delta h \delta ty_{i,j} * \phi_{i,j-1,src}^{t-1}$; **end if** {down, -ve = inflow}

               **if** $\delta h \delta t \text{sum}_{Q_{[\text{in}]}} = 0$ and $\phi_{src}^{t-1} = 0$ **then**

                    $\phi_{i,j,src}^t \leftarrow 0.0$ {if there is no contribution or existing water from this source}

               **else**

                    $\phi_{i,j,src}^t \leftarrow (h_{Q_{[\text{out}]}}^t * \phi_{i,j,src}^{t-1}) + \delta h \delta t \text{sum}_{Q_{[\text{in}]}} / h_{i,j}^t$ {([src depth in cell] + [src depth added]) / [updated depth]}

               **end if**

          **end for**

      **end if**

  **end for**

---



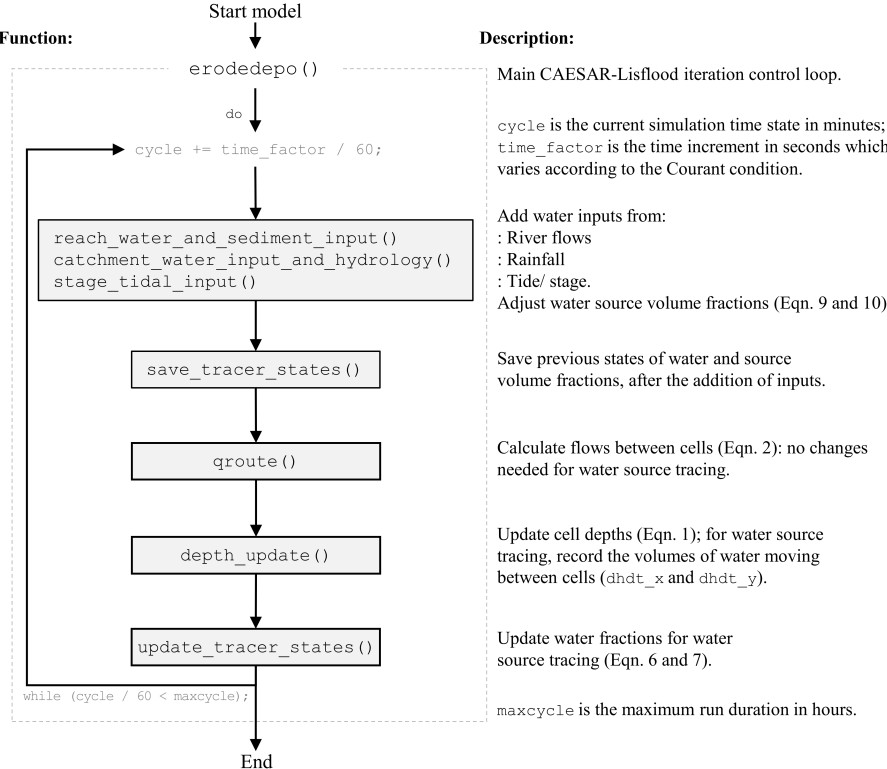

**Figure 1.** The main CAESAR-Lisflood control showing additions for water source tracing. Additional functions related to erosion and deposition are not shown.

The power term, $\beta$, enables visual enhancement of lower water fractions, in the range $\sim$0.1 to 1.0, where 1 would represent no enhancement. In order to allow water depth to be resolved through the use of reduced lightness for deeper water, (14) may be modified to:

$$RGB_{i,j} = ((1 - h_{i,j}/h_{[\text{range}]}) \cdot 127) + 128 \cdot \phi_{i,j,w}^{\beta} \qquad (15)$$

where $h_{i,j}$ is the cell depth and $h_{[\text{range}]}$ is the range of depths to scale the visualisation to. Thus, the RGB value for a cell which has only one source of water in it will range from 128 for deeper water where $h = h_{[\text{range}]}$, towards 255 for shallow depths close to zero. Figure 2 provides a detailed illustration of RGB values obtained using (14) and (15) for different values of $\phi$, $\beta$ and $h$, and the resulting colour scales obtained when visualising multiple sources. Mixing of water sources with red and blue shading gives a pink colour scale; green and blue gives cyan; red and green give yellow; and shades towards white would indicate mixing of all three sources. Note that, while it is possible to trace an arbitrary number of water sources using the method described in Section 2.2, this visualisation method is limited to a maximum of three water sources. However, using (15), if a cell only contains water from sources different to those selected for visualisation, depths can still be resolved in



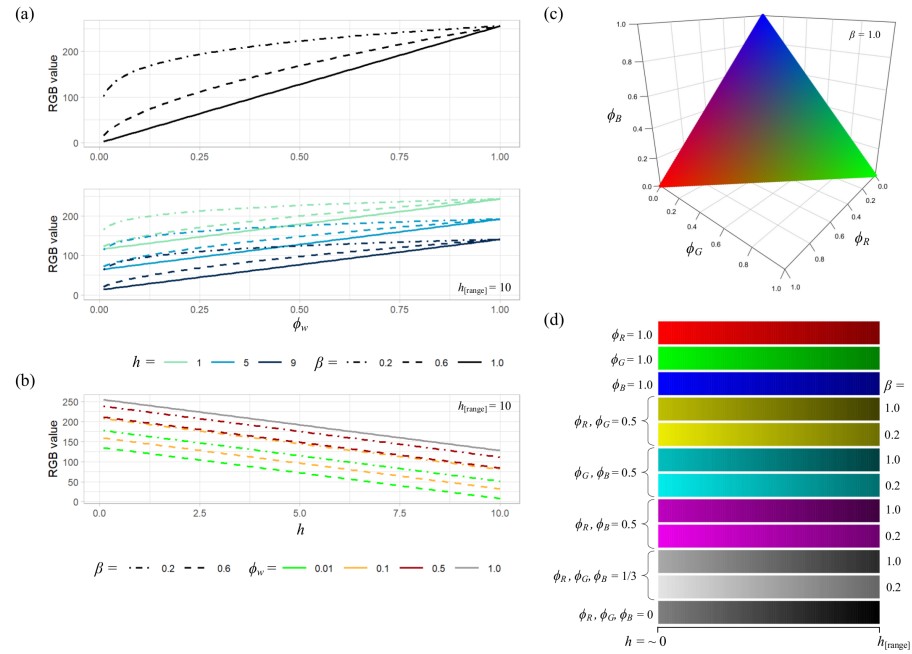

**Figure 2.** Illustration of RGB colour scaling used for visualisation of water source fractions: (a) RGB values for water fractions $\phi$ from source $w$, obtained using (14) (upper plot, no depth visualisation) and (15) (lower plot, with depth visualisation, where $h_{[range]} = 10$), and visual enhancement of lower water fractions using $\beta$; (b) RGB values with changing depth, $h$, for different values of $\phi_w$ and $\beta$; (c) colour rendering resulting from (14) for three water sources, $R$, $G$ and $B$ (note that $\phi_R + \phi_G + \phi_B = 1.0$); (d) colour scales obtained for various combinations of water sources using (15), with and without enhancement of lower water fractions, showing reduced lightness as water depth approaches $h_{[range]}$. Note that the bottom colour scale is used for rendering in the absence of any water from sources selected for visualisation (i.e. where more than three sources are present in the simulation).

greyscale in the RGB value range 0 to 127. This also means that it is possible to visualise only one water source at a time in a
selected primary colour.

## 2.5 Evaluating performance overhead

In order to test computational performance of the water tracing code, a simple planar test case was developed, consisting of a
constant 0.001 m/m slope of length 2000 m and width 1000 m, on a grid with a spatial resolution of 5 m, giving a total of
80,000 cells (Figure 3a). Manning's roughness, $n$, was set at 0.05. At 250 m intervals down slope, 1 m "walls" were added
across the slope, each with between 4 and 8 gaps of 5 m width added to allow water to flow through. This was done to
ensure that water mixed as it flowed downslope. Simulations were conducted for a period of 24-hours with constant inflow of
10 m³/s to 8 locations at the top of the slope. Eight simulations were conducted in total: one with no tracing and seven with
between 2 and 8 water sources being traced. To ensure that each of these simulations had identical total inflow, sources were



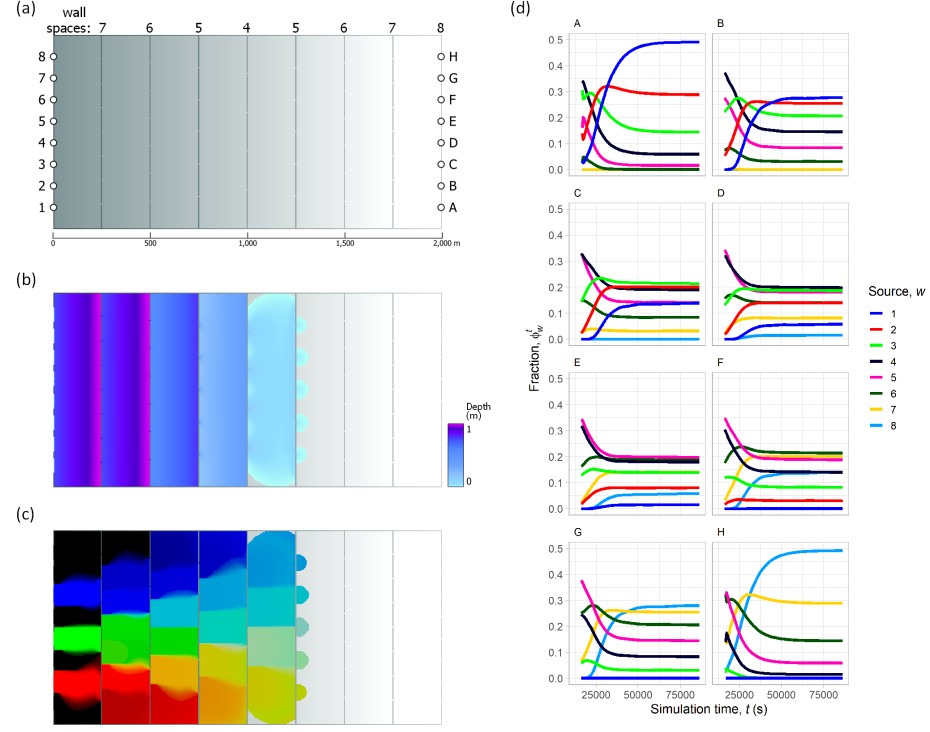

**Figure 3.** Planar test case setup and results: (a) the 2000 by 1000 m planar slope (0.001 m/m) elevation model (5 m grid spacing, 400 x 200 cells), with 1 m "walls" added across the slope at 250 m intervals (each wall had between 4 and 8 evenly-spaced gaps of 5 m, as indicated by the numbering along the top of the grid); injection points for each of the 8 water sources are indicated by numbering on the left; (b) water depth prediction at $t = 8400$ s (140 minutes), with a constant inflow of 10 m$^3$/s and $n = 0.05$; (c) visualisation of water sources numbered 2 (red), 4 (green) and 6 (blue) at $t = 8400$ s, using (14) with $\beta = 0.2$; (d) water source fractions throughout the 24-hour simulation for each of 1-8 sources at locations A-H indicated in (a). For an animated version, please see: https://youtu.be/DTw8ysJtx8o.

grouped for tracing purposes, allowing the variation in computation requirements to be assessed. The simulations conducted
are summarised in Table 1. Furthermore, to benchmark model performance in a real world scenario, the effect of the number
of tracers on model performance was tested on the Carlisle example described below.

## 3 Results

Firstly, we demonstrate water source tracing and visualisation for three flood inundation case studies that are examples of
flooding at three different spatial scales in three contrasting contexts: (1) a major flood event in 2005 at Carlisle, United King-
dom, fluvial flooding in an urban area which is situated at the confluence of three rivers, (2) a shallow estuary in Christchurch,
New Zealand, combining tides with inflow from two small rivers, and (3) flooding at the tributary junction of two large rivers

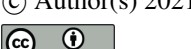



**Table 1.** Planar slope test case simulations. In each of the eight simulations, the total volume of inflow was identical: each of the eight sources was a constant 10 m$^3$/s (where a source was used for multiple inflow points, its total volume was added to all). Inflow points are shown in Figure 3a.

| Number of tracers | Inflow point/ source number | | | | | | | | Computation cost (s/s) | | Total computation time (s) | Difference to no tracing (s) | Ratio to no tracing |
|---|---|---|---|---|---|---|---|---|---|---|---|---|---|
| | 1 | 2 | 3 | 4 | 5 | 6 | 7 | 8 | Mean | Std. dev. | | | |
| No tracing | | | | 1 | | | | | 9.27E-03 | 2.51E-03 | 800.63 | - | - |
| 2 | 1 | | | | 2 | | | | 15.8E-03 | 4.38E-03 | 1362.46 | 561.83 | 1.70 |
| 3 | 1 | 2 | | | | 3 | | | 16.8E-03 | 4.70E-03 | 1452.47 | 651.84 | 1.81 |
| 4 | 1 | 2 | 3 | | | | 4 | | 18.0E-03 | 5.04E-03 | 1558.91 | 758.28 | 1.95 |
| 5 | 1 | 2 | 3 | 4 | | | 5 | | 19.2E-03 | 5.48E-03 | 1659.41 | 858.78 | 2.07 |
| 6 | 1 | 2 | 3 | 4 | 5 | | | 6 | 20.3E-03 | 5.90E-03 | 1750.80 | 950.17 | 2.19 |
| 7 | 1 | 2 | 3 | 4 | 5 | 6 | | 7 | 20.8E-03 | 6.13E-03 | 1799.12 | 998.49 | 2.25 |
| 8 | 1 | 2 | 3 | 4 | 5 | 6 | 7 | 8 | 22.3E-03 | 6.58E-03 | 1925.20 | 1124.57 | 2.40 |

on the Amazon floodplain, Brazil. Secondly, we present the computational performance of the method for the planar test case and Carlisle example.

### 3.1 Carlisle, United Kingdom

The city of Carlisle in northern England experienced significant flooding in 2005 (estimated annual exceedance probability 0.57% to 0.5%) that resulted from heavy rains in the headwater catchments of rivers running through the city and affecting more than 2500 properties (Environment Agency, 2006). With the benefit of an extensive set of observational data obtained from field collection, the 2005 flood event at Carlisle event has been used extensively as a test case for hydraulic model development (Horritt et al., 2010; Neal et al., 2009). Here, the site is of particular interest as it is at the confluence of three

separate rivers: the main River Eden, which runs from east to west through the city, is joined by the Rivers Petteril and Caldew which flow into the city from the south (Figure 4).

A simulation was run using a model grid of 5 m (domain size: 951 x 612 cells, 14.6 km$^2$), topography from LiDAR and inflows from gauging station records. The simulation began on 08 January 2005, 00:00 AM, for 120 hours (5 days). Visualisations of the model output are shown in Figure 5. Flood water mixing is shown in RGB colour space, using (15) to

enable darker shades to indicate deeper water. The mixing of Eden (blue) and Petteril (red) gives pink shades, which as it moves downstream transitions towards purple due to the larger volume of flow on the Eden; water from the Caldew is shaded in green and contributes to the greyer shades in the floodplains close to the main channel as the three waters are mixed.



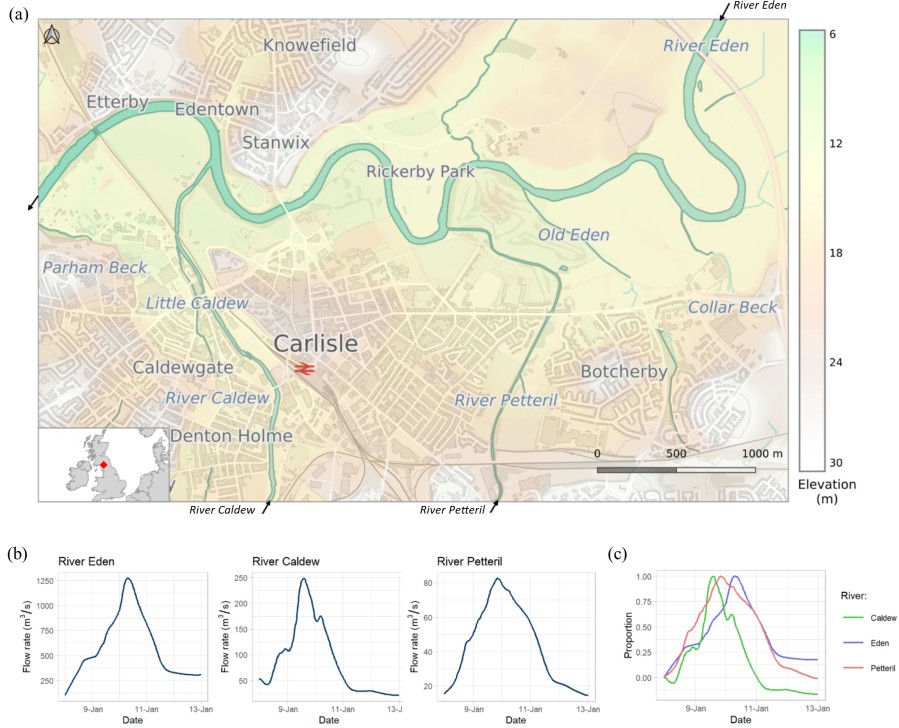

**Figure 4.** (a) Study site 1 at Carlisle, UK, at the confluence of the Rivers Caldew, Petteril and Eden, showing elevation from LiDAR. The plots in (b) show river flows for the January 2005 flood event simulated. The plot in (c) shows the relative level of the three rivers (calculated as a proportion of the difference between the flow on 8 January and the flood peak), indicates that the River Caldew reached flood peak ∼5.5 hours ahead of the River Petteril, and ∼17 hours ahead of the River Eden. Contains OS data © Crown copyright and database right (2020).

The ability to trace inputs from discrete tributaries allows us to determine how different sourced flood waters contributed to the flooding. This is especially pertinent to the Carlisle 2005 flood events, where most flood damage was considered to have

come from the two tributary inputs, the Caldew and Petteril (Environment Agency, 2006). For example, the time-series plots in Figure 5 illustrate locations where water is mixed from multiple sources. The larger volume of water from the River Eden dominates, but at point A, a railway embankment prevents flooding from the Eden but the area is flooded by the Caldew. Points B and D are initially flooded by the Caldew and Petteril, respectively, until flooding from the Eden arrives. This is likely due to the timing of the flood peaks (∼11.5 hours earlier on the Petteril than the Eden; ∼17 hours earlier on the Caldew).

**3.2    Avon-Heathcote estuary, Christchurch, New Zealand**

On the eastern edge of the city of Christchurch, New Zealand, the estuary of the Avon and Heathcote Rivers (Figure 6) is a large (∼8 km²), shallow (mean depth ∼1.4 m at spring high tide and a tidal range of ∼1.7-2.2 m) and intertidal area, with about 85% of its elevation above the spring low tide level (Findlay and Kirk, 1988). The estuary is semi-enclosed and separated from the Pacific Ocean by a ∼4.5 km long spit (Christchurch suburbs of New Brighton/ Southshore), has been designated



**Figure 5.** Model results for Carlisle, showing flood extent on 10 January 2005, 06:00 AM, close to the flood peak. Map colours indicate mixing of water, with $\beta = 0.2$ to emphasise lower water fractions from the Caldew and Petteril Rivers. Depths and fractions for four selected locations, marked A-D, are shown. Contains OS data © Crown copyright and database right (2020). For an animated version, please see: https://youtu.be/xOtOi06cXvA.



as a site of ecological significance in the Christchurch District Plan (Christchurch City Council, 2016) and is internationally recognised for its importance as a habitat for migratory bird species (Woodley, 2018). The estuary acts as a sediment trap for inflows from the Avon and Heathcote Rivers, the catchments of which cover a combined area of 188 $km^2$ which is largely urbanised, meaning that urban pollution is a considerable issue (Vopel et al., 2012).

Here, we demonstrate water source tracing and visualisation for the Avon-Heathcote estuary for the month of July, 2017,
using a model grid of 10 m (domain size: 483 x 675 cells, 32.6 $km^2$). The sources of water included were the two rivers, with inflow from the Avon River from the north and inflow from the Heathcote from the south-west, and a downstream tide boundary condition at the estuary outlet at the south-east (Figure 6b). The simulated period of July 2017 included both neap and spring tides and a high flow event on 22 July on both rivers, where the Avon River peaked at 13:00 with 14.7 $m^3/s$ of flow, close to the mean annual flood level (LAWA), and Heathcote River peaked at 14:00 with 28.1 $m^3/s$ of flow, an Annual
Exceedance Probability of less than 0.1 (LAWA).

As noted in Section 2.3, as a depth-averaged model, CAESAR-Lisflood does not account for the higher density of the saline tidal water compared to fresh riverine water, meaning that any variations in the vertical salinity profile of estuarine water is not accounted for. However, given that the Avon-Heathcote estuary receives only small volumes of riverine inflow, with mean inflow from the Avon and Heathcote rivers 1.87 and 1.04 $m^3/s$, respectively (LAWA), compared with tidal inflow rates of up
to 500-800 $m^3/s$ during an incoming tide (van der Peet and Measures, 2015), it is likely that it is well-mixed with little vertical salinity difference (Hansen and Rattray, 1966).

Results of the model application are shown in Figure 7. During low flow conditions during at low-tide on 8 July 2017 (image A), the estuary is mostly drained, with the water remaining in the estuary containing a greater proportion of water from the Avon River as a result of its higher baseflow. Shortly after spring-high tide on 16 July (image B), it can be seen that water
from both the Avon and Heathcote is pushed back by the tide, particularly along the eastern shoreline. If accurate, this situation may play a significant role in determining where pollutants, such as those from stormwater runoff which are then discharged from the Avon and Heathcote Rivers (e.g., see Vopel et al., 2012) are deposited. This would affect human health through various activities in the estuary, such as the gathering of food sources such as fish and shellfish which accumulate heavy metals contained in riverine discharge (e.g. McMurtrie, 2015).

During the high flow conditions on 22 July shortly after high tide and inflow peak (image C) and low tide (image D), results indicate substantially greater riverine contribution to the estuary, such that, after the fall of the tide, the estuary remained substantially inundated but with water almost all from riverine sources. After high tide, Avon and Heathcote waters are largely confined by the tide to the western shore, with a clear boundary between the two river sources visible. After low tide, this boundary is sharper due to the absence of tidal mixing, and gradually dissipates as water mixes towards the estuary outlet.

## 3.3 Amazon River, Brazil

The flow of the central Amazon River is characterized by a large annual flood wave of around 10 m amplitude, resulting primarily from the distinctive wet and dry seasons (Trigg et al., 2009). Each year during the wet season, the Amazon rises and inundates its low lying floodplain forests, creating an internationally significant wetland ecosystem which reaches its maximum





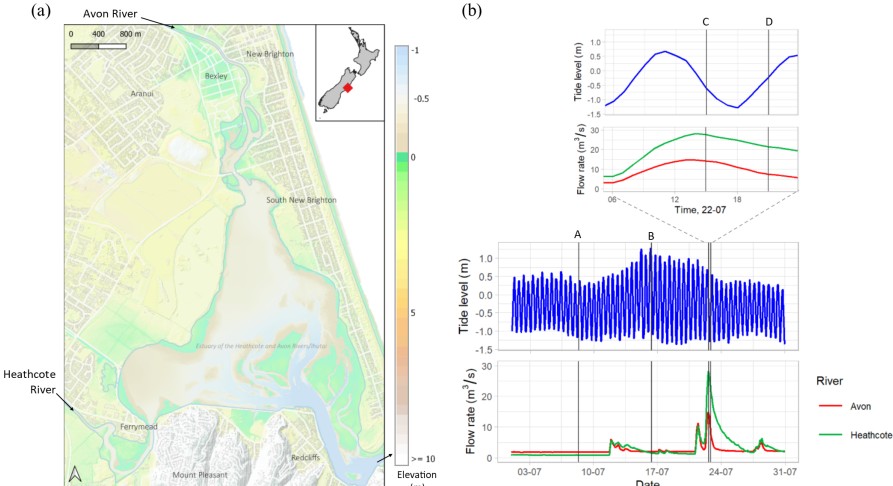

**Figure 6.** (a) Study site 2 at the Avon-Heathcote estuary in Christchurch, New Zealand showing elevation from LiDAR; (b) inflow and downstream tide levels for the period simulated in July 2017, which includes a high-flow event on 22 July. The four vertical lines labelled A-D represent the timing of the images shown in Figure 7. Tide data are from the NIWA Sumner Head gauge, around 1 km from the estuary outlet; flow data are from the Environment Canterbury gauges at Gloucester St. (Avon) and Buxton Terrace (Heathcote). Contains data sourced from the LINZ Data Service licensed for reuse under CC BY 4.0

extent around June or July (Mitsch and Gosselink, 2015), with the extensive surface water flow recognised a key factor in the
functioning of the habitats created and exerting a strong control on biological and biogeochemical processes (Richey et al.,
2002; Wittmann et al., 2004). Inter-annual variability in the flood level and flood wave timings creates spatial and temporal
complexities on the floodplain (Melack and Hess, 2011), with the source of water recognised as a key determinant for sediment
transport (Vauchel et al., 2017) and trace element concentration (Baronas et al., 2017) dynamics across the basin.

We applied the model of Wilson et al. (2007) to a ∼300 km reach of the Solimões River (mainstem Amazon River) at its
confluence with the Purus River (Figure 8), using a model grid of ∼278 m (domain size: 966 x 479 cells, ∼35,650 km$^2$) for the
period of October 2013 through December 2014. Operating at a far larger spatial scale to examples 1 and 2, here water source
tracing results show that, downstream (west) of the river confluence, the southern floodplain is primarily inundated with water
from the River Purus (indicated by green), while the floodplain to the north contains a mix of water from the Rivers Solimões
and the Purus (indicated by orange) (Figure 9). The shared floodplain upstream of the confluence displays a clear gradient
transition between the two water sources, passing from red (Solimões), through yellow (equally mixed) to green (Purus). The
cross-section profile plots in Figure 9 show that, after flowing into the Solimões, water from the Purus remains largely on the
right (southern) bank, but mixes with the water from the Solimões as the river progresses downstream. As the Purus peaks
sooner than the Solimões, the proportion of water from the Purus close to the right bank decreases through the Solimões flood
peak (C) and mid-falling water (D). These demonstrate the ability of the LISFLOOD-FP and the new tracing functions to
simulate the convergence and mixing of water from these two major river systems. The proposed water source tracing method



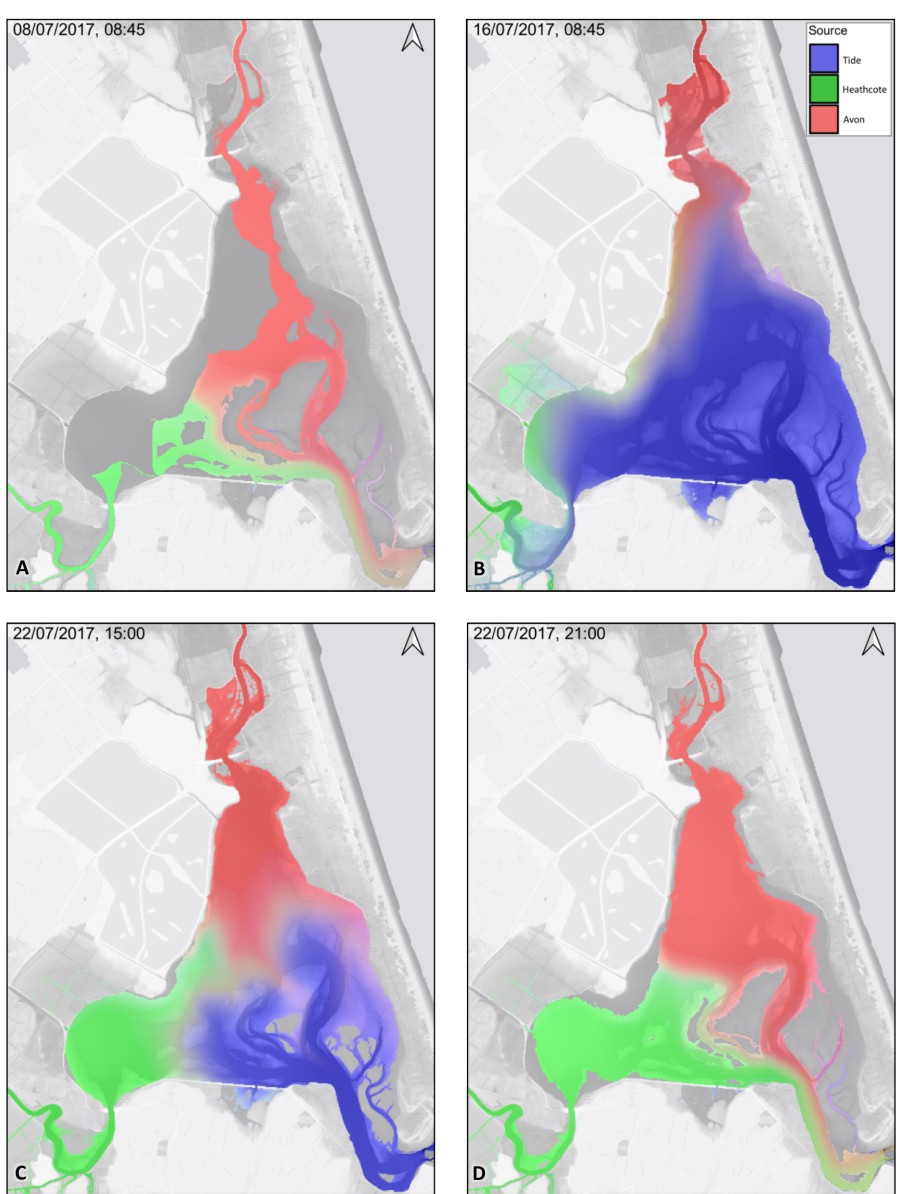

**Figure 7.** Model results for the Avon-Heathcote estuary: low flow conditions during (A) low-tide on 8 July 2017 and (B) after spring-high tide on 16 July; high flow conditions on 22 July shortly after high tide (C) and low tide (D). The images were produced with $\beta = 0.2$; their timing and flow/ tide levels are shown in Figure 6. The background images contain the NationalMap Basemap (CC-BY-ND license) and LiDAR data shaded in grey. For an animated version, please see: https://youtu.be/Fczr5tczzXU.



therefore provides the means to assess the complex dynamics of variable contributions in river-floodplain systems such as those on the Amazon, enabling the estimation of important variables which are associated with each water source through time, such as nutrient availability.

### 3.4 Computational performance

Results of the planar test case are shown in Figure 3b-d. Flow depth at $t = 8400$ s is shown in Figure 3b, and a water source visualisation at the same time for the simulation with all 8 sources is shown in Figure 3c. Here, flood water mixing is shown in RGB colour space, as per (14) i.e. the mixing of sources 2 (red) and 4 (green) gives yellow; mixing source 4 and 6 (blue) gives cyan. Other water sources are shown in black. As the water flows down the slope, it becomes increasingly mixed within each section, helped by water being forced to flow through the gaps left in the walls placed across the slope. Once water flow reaches 260 the bottom of the slope, a high proportion of the domain contains water (77,901 out of 80,000 cells). The flow is constrained to flow out of the domain through eight gaps in the downstream boundary wall, marked A-H in Figure 3a. The fractions of water sources 1-8, at locations A-H, throughout the simulation are shown in Figure 3d. As the domain has mirror symmetry along the west to east axis at $y = 500$ m, the fractions at each of the locations are also symmetrical with respect to water sources. For example, at location A, water from source 1 increases to a fraction of around 0.5 of the water flowing through this location at 265 steady-state, while water from source 8 does not reach it at all; the reverse is true for these water sources at location H. For the two central locations, D and E, all water sources are present with steady-state fractions ranging from around 0.02 (for source 8 at location D or source 1 at location E), to 0.2 (for sources 4 and 5).

Code profiling using diagnostic tools of Microsoft® Visual Studio 2019 compiler software (version 16.11.2), based on this test application with 3 sources, indicates that `update_tracer_states()` uses around 17.9% of the CPU resources 270 allocated to CAESAR-Lisflood during a simulation, with 3.0% used for `save_tracer_states()`. This compares to 39.6% for `qroute()` and 11.5% for `depth_update()`. This profiling was completed on a Microsoft® Windows® 10 (build 18363) computer system with an Intel® Xeon® E-2278G CPU running at 3.4 GHz base (5.0 GHz max. turbo) frequency with 8 cores/ 16 threads, 128 Gb of system memory (4 x 32 Gb Samsung M391A4G43MB1-CTD ECC UDIMM running at 2,666 Mbps), writing outputs to a Samsung NVMe Solid State Disk (MZVLB1T0). All simulations presented in this paper 275 were completed on this computer system.

The total time of each of the eight simulations is presented in Table 1, ranging from 1362.5 s with 2 tracers, up to 1925.2 s with 8 tracers, as compared to 800.6 s without water source tracing. The ratio of computation time for the full simulations with tracing, against computation time without tracing, ranges from 1.7 for two sources to 2.4 for eight. Mean computational requirements time ranged from 0.016 s/s for 2 tracers to 0.022 s/s with 8 tracers, compared to 0.009 s/s without water source 280 tracing. In addition, the variability in computational costs increases with the number of sources being traced, likely due to the complexity of the mixing introduced by constraining flow through gaps within this test case.

Figure 10 illustrates the computational efficiency of our scheme for the Carlisle flood model, with and without water source tracing. Computational requirements for the simulation with water source tracing were found to be between 1.2x and 1.5x the simulation without tracing, which is less than the 1.8 ratio found for the planar test case with 3 water sources. As with the





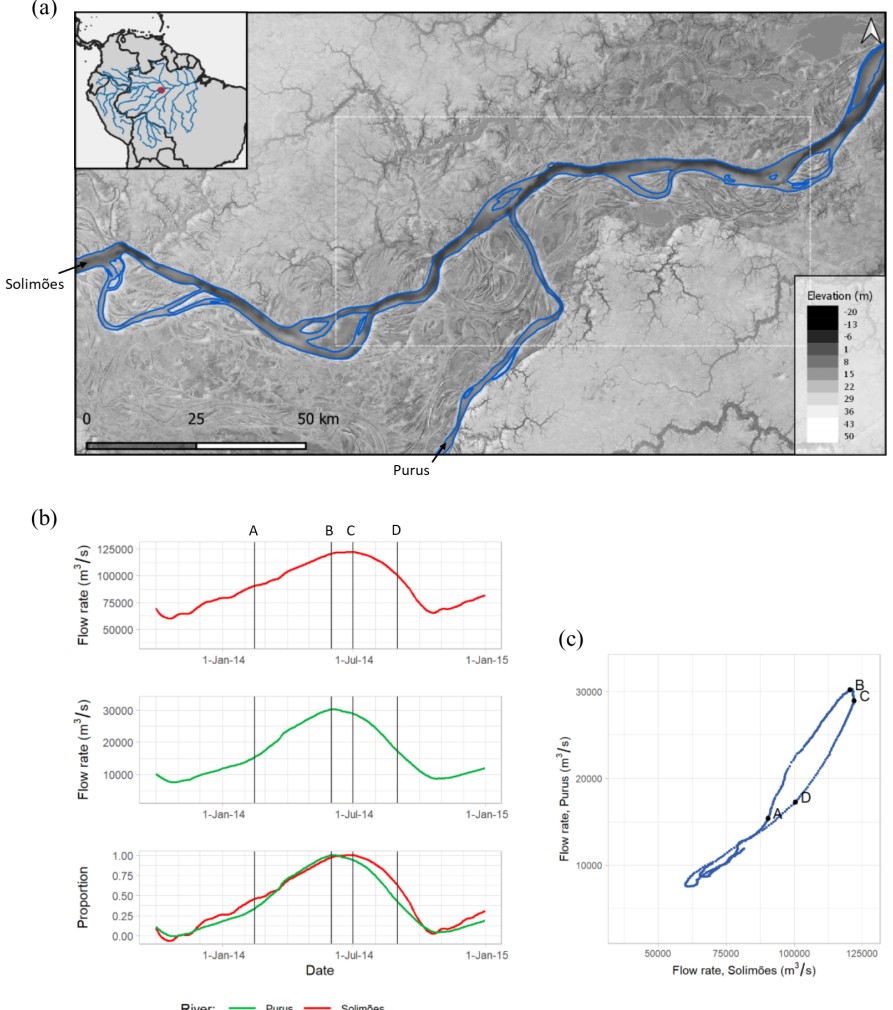

**Figure 8.** (a) Study site 3 at the confluence of the Solimões (mainstem Amazon) and Purus rivers in the central Amazon, Brazil, showing elevation derived from SRTM (O'Loughlin et al., 2016) and bathymetry (Wilson et al., 2007). The plots in (b) and (c) show river flows for the period simulated from October 2013 through December 2014. The lower plot of (b), which shows the relative level of Solimões and Purus flow (calculated as a proportion of the difference between the flow on 1 November 2013 and the 2014 peak flow), indicates that the Solimões rises more steadily and before the Purus, which has a later, steep rise from around mid-February onwards. Peak flow levels on the Purus (indicated by letter B) occurred around one month earlier than peak flow on the Solimões. The dotted box in (a) and letters A-D in (b) and (c) indicate the location and timing of the model results shown in Figure 9. Contains Natural Earth data in the public domain.



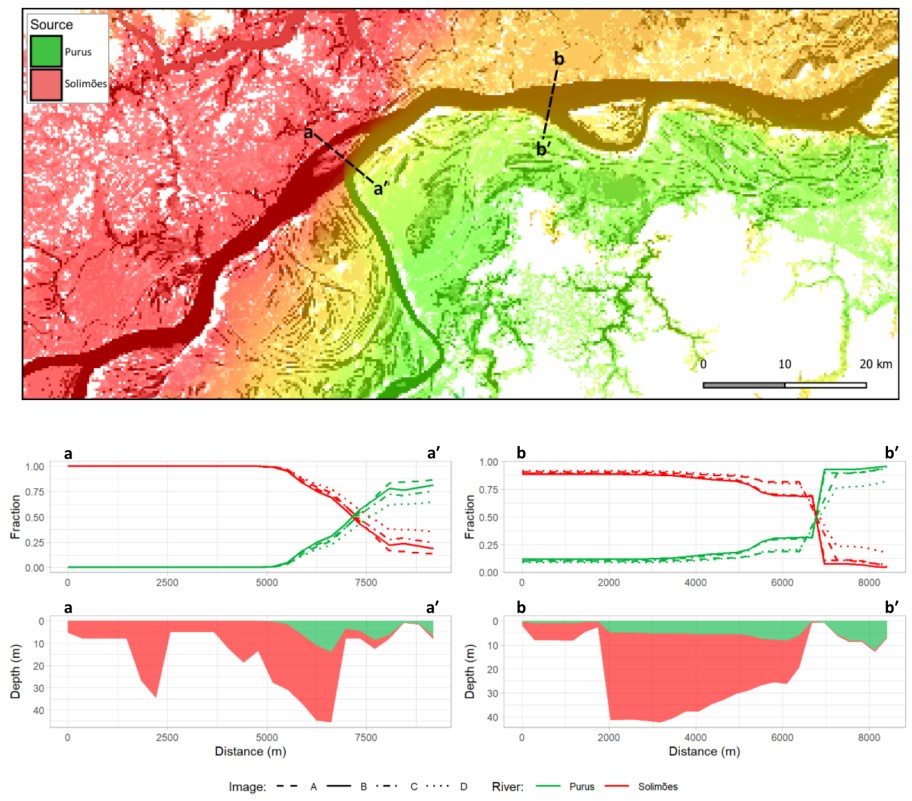

**Figure 9.** Model results for the Amazon, with colours on the map at the Purus flood peak (top) indicating the mixing of the waters along the channel and across the floodplain. Channel cross-section plots (bottom) at locations indicated by a-a′ and b-b′ show the water source fractions (upper plots) for each channel (A = mid-rising; B = Purus flood peak; C = Solimões flood peak; D = mid-falling), and depth/ source profiles (lower plots) for image B (Purus flood peak). For an animated version, please see: https://youtu.be/PknAL_8fd1I.

planar test case, the computational requirements of water source tracing were found to increase with the area of inundation. Early in the simulation, when the flood area was around 1 $\text{km}^2$ (40,000 cells), the computational overhead of water source tracing was approximately 0.01 s/s. This increased nearly linearly to ~0.05 s/s as the flood extent reached its peak extent of around 5 $\text{km}^2$ (200,000 cells).

## 4 Discussion

Our method for water tracing is simple, effective and has a low additional computational overhead of 1.2-1.5 for our real world test cases. The three case studies above show that the approach is flexible, applicable in urban, coastal and fluvial environments over a wide range of spatial scales. An important advantage of the method is that the number or water sources which may be traced is limited only by computational constraints. Furthermore, as the formulation records the proportion of different tracers



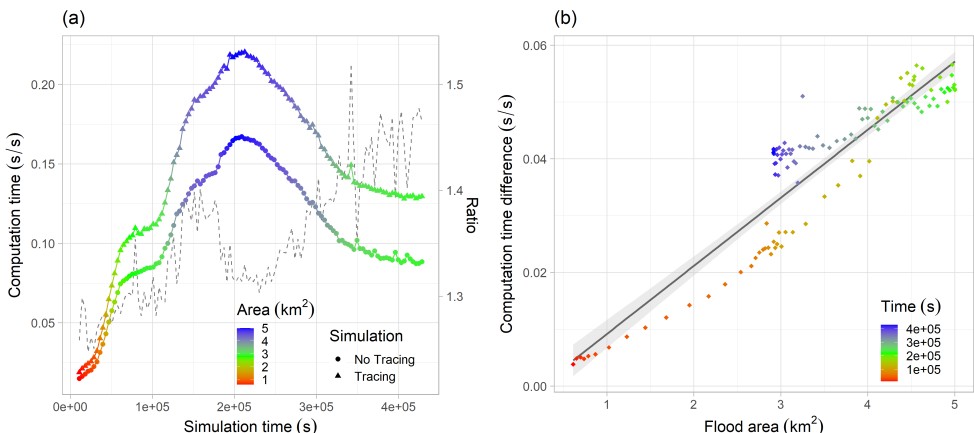

**Figure 10.** Computational efficiency of water source tracing for the Carlisle simulations with and without water source tracing: (a) computation time required per second of the 5-day (432,000 s) simulation, with colours indicating the area of inundation; the dashed line shows the ratio between the two (secondary y-axis); (b) computation time difference (i.e. cost of tracing) against flood area, with a linear model fitted for reference; colours indicate the simulation time.

or water sources per cell, it is fully independent of the hydraulic calculations, meaning it is straightforward to add to existing

finite volume codes and similar 2D flow models to LISFLOOD-FP. This also means that predicted water levels with or without the tracers are identical.

However, it is important to remember that the speed and simplicity of the method is at the expense of treating cell water volumes as being fully mixed. This means that it is possible for small fractions of a tracer/water from different sources to propagate quickly downstream, since fluxes into a cell would be included for **all** the fractions assigned as an inflow to a

downstream neighbouring cell. In effect, this represents a tiny amount of numerical diffusion and the volumes moved are very small, but as a result caution should be applied to the interpretation of very small water source fractions. This is also an issue for tracing functions in other codes such as TELEMAC (e.g. Ch. 9.5 Ata et al., 2014). Additionally, as each cell is fully mixed we cannot yet incorporate the effects of different water densities, such as saline water.

These issues notwithstanding, this method provides excellent opportunities for simplified simulation of water, solute and wa-

ter borne fluxes across river catchments and in estuaries. In particular, with the CAESAR-Lisflood implementation combining a hydrological and hydraulic model, the contributing effects of sub basins on flooding can easily be simulated. Furthermore, the ability to combine both point source (e.g. from combined sewer outflows) and diffuse source (e.g. fields) contaminant sources within a computationally fast model opens up many opportunities to simulate water quality and pathogen/contaminant issues.



## 5   Conclusions

We developed a simple method for tracing water sources through a flood inundation model and demonstrated its application
for a simple test case and three example case studies at different spatial and temporal scales. A key advantage of the approach
lies in its independence from the hydraulic methodology used, meaning that it is relatively straightforward to add to existing
in finite volume codes. The method enables effective and informative visualisation of flood inundation, providing additional
insight into flood dynamics. In addition, it may potentially provide the ability to simulate the movement of solute contaminants

and pathogens within a computationally efficient model, although additional testing is required to verify the reliability of the
method, given the assumption of full mixing within cells.

*Code and data availability.*   The CAESAR-Lisflood v1.8f source code with added water source tracing is freely available for download
from Zenodo under the GNU General Public License v3.0, along with the planar test case and a 15 m version of the Carlisle model at
https://doi.org/10.5281/zenodo.5541123 (Wilson and Coulthard, 2021a).

*Video supplement.*   Animations for each of the case studies are available at the YouTube links provided in the manuscript, and from Zenodo
at https://doi.org/10.5281/zenodo.5548535 (Wilson and Coulthard, 2021b).

## Appendix A:  Code description

The tracing component requires a series of additional variables representing the water state prior to the routing (Figure A1).
The variable `water_depth_prev` is an array of $X * Y$ grid cells containing the depth from the previous time step; `dhdt_x`

and `dhdt_y` are arrays containing the change in cell depth in the last time step in for columns and rows, respectively. The
fraction of each cell from each source (i.e. the water tracers) are recorded in a series of three-dimensional ($W * X * Y$
grid cells) or stacked arrays containing the tracer fractions per cell for present and previous iterations (`watertracer`,
`watertracer_prev`) the rain zonations (`watertracerRainZone`, `watertracerRainZone_prev`) and a series
of flags to enable or disable tracing. Note that the rainfall source as a whole (i.e. from any zone) is contained within the main

`watertracer` arrays, while the fraction of rainfall from each zone is recorded in the `watertracerRainZone` arrays.

```
1  public static double[,] water_depth_prev, dhdt_x, dhdt_y; //water state prior to routing
2  public static double[, ,] watertracer, watertracer_prev, watertracerRainZone, watertracerRainZone_prev;
3                                        // stacked arrays for tracing water sources
public static int[,] rainzonation; // rainfall zonation for routing
public static int nSources = 0, nRainZones = 0; //number of water sources for tracing, rain zones
bool isTraceWater = false; // Water source tracing (default to off)
bool isTraceRainZonation = false; // Water source tracing: rainfall zonation
public int[] sourceIDs; // to identify different sources where they are split over several cells
public static int[] rainZones; // list of rainfall zones
```

**Figure A1.** Additional variables required for water source tracing.





The addition to the reach inputs is shown in Figure A2, from the function `reach_water_and_sediment_input()` (rainfall inputs and tidal or stage inputs are added similarly but code is omitted for brevity). In this function, the updated water source fractions for the water source being added to the cell (Eqn. 9) and other sources already in the cell (Eqn. 10) are implemented on lines 33 and 40, respectively.

```
for (int n = 0; n <= number_of_points - 1; n++)
{
int x = inpoints[n, 0];
int y = inpoints[n, 1];
double interpolated_input1 = inputfile[n, (int)(cycle / input_time_step), 1];
double interpolated_input2 = inputfile[n, (int)(cycle / input_time_step) + 1, 1];
double proportion_between_time1and2 =
((((int)(cycle / input_time_step)+ 1 ) * input_time_step) - cycle) / input_time_step;
double input = interpolated_input1 +
((interpolated_input2 - interpolated_input1) * (1-proportion_between_time1and2));
double dhdt = 0.0, h_from_this_source = 0.0, prev_depth = 0.0; // Needed for new water tracing fractions
prev_depth = water_depth[x, y]; // Save previous depth prior to updating
dhdt = (input / div_inputs) / (DX * DX) * local_time_factor; // Additional depth from this source
water_depth[x, y] += dhdt; // Update depth for cell x,y
if ((isTraceWater == true) && (water_depth[x, y] > 0.0))
{
int thissrc = sourceIDs[n + 1] + 2; // identify the source of this input
if (water_depth[x, y] == dhdt) // if depth was zero, all water in this cell is from this input
{
watertracer[x, y, thissrc] = 1.0; // this source layer
}
else if (watertracer[x, y, thissrc] < 1.0) // if all water is already from this source (i.e. == 1.0), no
27                                                // need to update
{
h_from_this_source = watertracer[x, y, thissrc] * prev_depth; // amount of water from this source
30                                                                     // already in cell
32       // update this layer water fraction - i.e. increase
watertracer[x, y, thissrc] = (h_from_this_source + dhdt) / water_depth[x, y];              // Eqn. 9
35       // update other layers water fraction - i.e. reduce
for (int src = 1; src <= nSources; src++)
{
if (src != thissrc) // skip this source layer
{
watertracer[x, y, src] = (watertracer[x, y, src] * prev_depth) / water_depth[x, y];       // Eqn. 10
}
}
}
}
}
```

**Figure A2.** Updating cell water fractions according to (9) and (10), from the function `reach_water_and_sediment_input()`.

335    After the addition of boundary condition inputs from each source, the new tracer states (prior to surface water routing) are saved within the function `save_tracer_states()`, in which the `waterdepth`, `watertracer`, and `watertracerRainZone` arrays are copied into the `waterdepth_prev`, `watertracer_prev` and `watertracerRainZone_prev` arrays, respectively (Figure A3).

Flow rates between all cells are then calculated as normal in the function `qroute()` (i.e. solving Eqn. 2 between all cells),
340    with no modifications needed to the code for water source tracing. Once flows are calculated, depths are updated according to (1) in the function `depth_update()`. Here, the only modification required for water source tracing is that the volumes of water moving between cells is recorded in the `dhdt_x` and `dhdt_y` arrays (Figure A4). Here, the `dhdt_x` and `dhdt_y` arrays are updated on lines 18-24, before the `water_depth` array is updated according to (1) on line 27. For brevity, the code presented in Figure A4 omits some additional processing for updates to suspended sediment and error checking.




```
1   void save_tracer_states()
2   {
3     // Save previous states of water and source fractions, after the addition of inputs
var options = new ParallelOptions { MaxDegreeOfParallelism = Environment.ProcessorCount * 4 };
Parallel.For(1, ymax + 1, options, delegate(int y)
{
int inc = 1;
while (down_scan[y, inc] > 0)
{
int x = down_scan[y, inc];
inc++;
water_depth_prev[x, y] = water_depth[x, y];
for (int z = 1; z <= nSources; z++)
{
watertracer_prev[x, y, z] = watertracer[x, y, z];
}
if (isTraceRainZonation == true)
{
for (int z = 1; z <= nRainZones; z++)
{
watertracerRainZone_prev[x, y, z] = watertracerRainZone[x, y, z];
}
}
}
});
}
```

**Figure A3.** `save_tracer_states()`

```
void depth_update()
{
double local_time_factor = time_factor;
if (local_time_factor > (courant_number * (DX / Math.Sqrt(gravity * (maxdepth)))))
local_time_factor = courant_number * (DX / Math.Sqrt(gravity * (maxdepth)));
6     // ...
var options = new ParallelOptions { MaxDegreeOfParallelism = Environment.ProcessorCount *  4 };
Parallel.For(1, ymax+1, options, delegate(int y)
{
int inc = 1;
12      // ...
while (down_scan[y, inc] > 0)
{
int x = down_scan[y, inc];
inc++;
if (isTraceWater == true)// for water tracing, record the volumes of water moving between cells
{
dhdt_x[x + 1, y] = local_time_factor * qx[x + 1, y] / DX;
dhdt_x[x, y] = local_time_factor * qx[x, y] / DX;
dhdt_y[x, y + 1] = local_time_factor * qy[x, y + 1] / DX;
dhdt_y[x, y] = local_time_factor * qy[x, y] / DX;
}
26        // update water depths
water_depth[x,y] += local_time_factor * (qx[x + 1, y] - qx[x, y] + qy[x, y + 1] - qy[x, y]) / DX;
29        // ...
}
31      // ...
});
33    // ...
}
```

**Figure A4.** Modified `depth_update()` function

Finally, the main addition to the CAESAR Lisflood code for water source tracing is within the function `update_tracer_states()`, that deals with the mixing component and updates the different source tracer fractions as water is moved between cells during each iteration (Figure A5). In the `update_tracer_states()` code, the cell depth after removal of outflows, $h^t_{Q_{[out]}}$ (7), is first calculated (lines 31-43). Then, the updated volume fractions (6) obtained for each source are calculated (lines 45-81), with the total volume contributed from each source first calculated (lines 47-58), then the updated cell volume fractions obtained





350 (lines 77-79). Finally, the function will update the volume fractions for each rainfall zone source in a similar way (code omitted for brevity).

*Author contributions.* MW designed the methods, developed the code and performed the model simulations. TC verified and validated the methods and code. MW and TC jointly prepared the manuscript.

*Competing interests.* The authors declare no competing interests.

355 *Acknowledgements.* An earlier version of this work was presented at the GeoComputation 2019 conference in Queenstown, New Zealand (Wilson and Coulthard, 2019). Research was co-funded by UKRI NERC grant numbers NE/K008668/1, NE/R009007/1 and NE/V004247/1. LiDAR elevation data and river flow data for the Carlisle site were provided by DEFRA/ the Environment Agency under the Open Government Licence. For the Avon-Heathcote site, LiDAR data and mapping data were provided by Land Information New Zealand (LINZ), river flow data were provided by Environment Canterbury, and tide data were provided by the National Institute for Water and Atmospheric

360 Research Ltd. (NIWA). For the Amazon site, Shuttle Radar Topography Mission were used, corrected for vegetation by (O'Loughlin et al., 2016), with additional bathymetry data from (Wilson et al., 2007); river flow and stage data were provided by the Agência Nacional de Águas (ANA), Brazil. We gratefully acknowledge Rob Thomas for his comments on an earlier version of the manuscript.





```csharp
1  void update_tracer_states()
2  {
3  // Update water fractions for water source tracing. Note - we only need to
4  // deal with inflows for each cell as it is assumed that the water in a cell is mixed,
5  // so the fraction from each source in outflow will be the same as in the cell itself.
6  //
7  // 1. Get depth after outflows only - check not to get -ve depths
8  // 2. Get dhdt added by each inflow and scale for each water source
9  // 3. In main cell, work out the sum of depth from each source:
10 //    : the fraction of remaining water from each source, plus
11 //    : the sum of dhdt from each source
12 // 4. Update the fractions from each source: divide by the new depth
var options = new ParallelOptions { MaxDegreeOfParallelism = Environment.ProcessorCount * 4 };
Parallel.For(1, ymax + 1, options, delegate(int y)
{
int inc = 1;
while (down_scan[y, inc] > 0)
{
int x = down_scan[y, inc];
inc++;
if (water_depth[x, y] > 0.0) // assess all cells which contain water
{
double dhdt_sumOut, depth_after_outflows, raindepth_after_outflows, raindepth;
double[] dhdt_sumInSrc;
dhdt_sumInSrc = new Double[nSources + 1];
double[] dhdt_sumInSrcRainZone = new Double[nRainZones + 1];
30       // work out total outflows for this cell - source not important here
dhdt_sumOut = 0.0;                                              // start Eqn. 7
if (dhdt_x[x + 1, y] < 0.0) {
dhdt_sumOut += dhdt_x[x + 1, y]; }  // Flow from right: -ve = outflow
if (dhdt_x[x, y] > 0.0) {
dhdt_sumOut -= dhdt_x[x, y]; }      // Flow from left:  +ve = outflow
if (dhdt_y[x, y + 1] < 0.0) {
dhdt_sumOut += dhdt_y[x, y + 1]; }  // Flow from up:   -ve = outflow
if (dhdt_y[x, y] > 0.0) {
dhdt_sumOut -= dhdt_y[x, y]; }      // Flow from down:  +ve = outflow
40       // n.b. sum of outflows will be negative
42       // get depth of cell from previous iteration after outflows
depth_after_outflows = water_depth_prev[x, y] + dhdt_sumOut;    // end Eqn. 7
for (int src = 1; src <= nSources; src++)                       // start Eqn. 6
{
dhdt_sumInSrc[src] = 0.0;
49         // work out the total inflow from neighbouring cells for this source of water:
50         // ignore outflows
if (dhdt_x[x + 1, y] > 0.0) { dhdt_sumInSrc[src] +=
dhdt_x[x + 1, y] * watertracer_prev[x + 1, y, src]; } // Flow from right: +ve = inflow
if (dhdt_x[x, y] < 0.0) { dhdt_sumInSrc[src] -=
dhdt_x[x, y] * watertracer_prev[x - 1, y, src]; } // Flow from left:  -ve = inflow
if (dhdt_y[x, y + 1] > 0.0) { dhdt_sumInSrc[src] +=
dhdt_y[x, y + 1] * watertracer_prev[x, y + 1, src]; } // Flow from up:   +ve = inflow
if (dhdt_y[x, y] < 0.0) { dhdt_sumInSrc[src] -=
dhdt_y[x, y] * watertracer_prev[x, y - 1, src]; } // Flow from down:  -ve = inflow
60         // update sources at this location
if ((dhdt_sumInSrc[src] == 0.0) & (watertracer_prev[x, y, src] == 0.0))
62         // if there is no contribution or existing water from this source
{
watertracer[x, y, src] = 0.0;
}
else // if there are inflows or existing water from this source, update fractions
{
if (depth_after_outflows < 0) // just in case
{
watertracer[x, y, src] = dhdt_sumInSrc[src] / (water_depth[x, y] -
depth_after_outflows); // fractions assigned based only on incoming water
}
else
{
75             // ([depth of water from this source still in cell] + [depth of water from this
76             //  source flowing in]) / [total depth now in cell]
watertracer[x, y, src] = ((depth_after_outflows * watertracer_prev[x, y, src]) +
dhdt_sumInSrc[src]) / water_depth[x, y];
}
}
}                                                              // end Eqn. 6
82       // deal with rain zones
if (isTraceRainZonation == true && watertracer[x, y, 1] > 0.0)
{
85         // ...
}
}
}
});
}
```

**Figure A5.** The main water source tracing calculations within the `update_tracer_states()` function



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
