# Peer review of "Tracing and visualisation of contributing water sources in the LISFLOOD-FP model of flood inundation (within CAESAR-Lisflood version 1.9j-WS)"

_Geoscientific Model Development, 2021_

## Referee Comment (RC1)

**A review of "Tracing and visualisation of contributing water sources in the LISFLOOD-FP model of flood inundation" (Wilson and Coulthard, preprint 2021)**

**Testing the model**

I followed the instructions on the Zenodo archive page (https://zenodo.org/record/5541123) to test the model on a Windows 10 laptop. It would be good to have a clearer step-by-step tutorial for the user depending on which case study they wanted to look at, just to make the page a bit more user-friendly. For example, I wasn't sure whether all the files on the archive were required for testing , but then I realised I would only need either the *CarlisleTestCase.zip* or the *PlanarTest.zip* file which would have all the files I needed.

The Carlisle test case worked out of the box, but the Planar test case required some editing of the XML file. It would be good to perhaps tweak the files within *PlanarTest.zip* so that it works out of the box like the Carlisle test case.

*Carlisle Test Case*

The data loaded in fine and a dialogue box showed that the variables were OK. I did not finish the simulation as it was still going after a few hours. But the in-progress map seemed to be approaching what was displayed in the paper.

*Planar Test Case*

This did not work for me out of the box. I opened the *CAESAR-lisflood 1.8_WS* application and opened the *model_trace_8.xml* file per instructions. When I clicked load data, this dialogue box appeared:

[Figure]

The folder for *PlanarTest* contained the *elevgrid.asc* file, so it is possible that the programme is not seeing it. The .xml file for the Carlisle test case did not have the trailing characters at the beginning of the file, so I edited the *model_trace_8.xml* file's DEM section to look like this:

[Figure]

When I clicked load data, this new dialogue box appeared:

[Figure]

Pressing OK led to a new dialogue box with the same message, but the integer in the filename was incremented by 1 each time (e.g. *hydro_steady_ch1.txt* until *hydro_steady_ch8.txt*). Following the same method used to fix the DEM filename, I edited the .XML file to remove the trailing characters "..\" at the beginning of each filename. After loading this in again, the dialogue boxes were gone and the programme told me "All other variables are OK" and let me run the model.

It did not look like anything was happening at first because the screen was not automatically updating as it had done in the Carlisle test case. But pressing the *update graphics* button caused the screen to update with the images expected. I am unsure why the Carlisle test case automatically generates the images, but the Planar test requires the user to press the *update graphics* button.

---

## Author Response (AR1)

**Author's responses (gmd-2021-340)**

Reviewers comment in black/ italics. Authors responses in blue

*Anonymous Referee #1*

*The paper presents an algorithm that allows water source tracing and visualisation of water sources for flood inundation. The algorithm and concepts are outlined and described clearly in the paper, and each step of the process is easily understood through the text. The uses and implications of the algorithm that involve contaminant tracing or how inundation water quality is affected by different sources is also briefly discussed but could have been further emphasised in the paper. Having three case studies of varying sizes, flood mechanics and origins, and timescales was good since it showed the algorithm's flexibility. Overall, the paper is well-written and the algorithm contributes further to the field of floodplain inundation modelling. I would recommend it for publication with minor revisions through adding some additional information to the introduction and discussion, and improving the presentation of some of the results.*

*Introduction*

*The LISFLOOD-FP model is briefly described in the introduction (purpose, use, representation of floodplain). The CAESAR model is also mentioned (P2, L33) but it does not get a similar description. It would be good to see a few lines describing the CAESAR model so the reader knows its purpose within the CAESAR-Lisflood model.*

We already have a 2-3 line description of the CAESAR-Lisflood model - with key references. Whilst a fuller explanation might be of interest to some readers, the unique aspects of CL (modelling morphodynamic changes etc...) are not relevant to this paper or demonstration of the tracing methods we have developed. Therefore we have left the text as is. If the editors think an additional description is required - we can readily add this.

*It would be good to see a brief review of previous work around tracing flood water sources in flood models in simpler schemes like LISFLOOD-FP. P1 L21-22 says that this ability is presently missing from reduced-complexity models, so have there been other papers that have tried to represent water source tracing in floodplain inundation models?*

To the authors knowledge, there are no other schemes for tracing water within the Lisflood-FP framework that has been widely adopted and modified for use in several commonly used flood models/codes. More complex (mathematically and computationally) models do have this function such as Telemac-3D that we mention in the following paragraph.

However, further to R2's comments we have also added a section detailing an additional hydraulic scheme that allows for water source tracing (L30-36):

> Similarly in PCSWMM, Qi et al. (2021) and Qi et al. (2022) developed a module to assess the relative contributions to downstream flood waters from upstream source catchments, with tracer sources generated by the PCSWMM water quality routing module. In their approach, each upstream source is assigned a constant tracer concentration, which is then routed downstream using PCSWMM methods for transportation of pollutants. Source proportion for a downstream

catchment is then determined from the relative mass of the total tracer amount which is in that catchment, multiplied by the total flood volume.

*Methods*

*P4 L88-89: "Thus, fractions from sources where water is added to the cell are adjusted upwards, while fractions for non-source volumes are adjusted downwards."*

*Why? What would be the physical basis behind this?*

This means that where concentrations of a certain source are increasing, this results in an increase in the fraction of that source in the water source fraction/proportion variables - and the opposite when concentrations decrease due to dilution from other sources. Text has been added to clarify this (L101-104).

*P5 L130-131: "… each of the four flow directions"*

*What are some of the advantages and limitations when four flow directions are considered? Would using the D8 or D-infinity representations of flow direction meaningfully affect the final results?*

There are advantages of D8 routing - in that for single cells channel widths water can pass diagonally across a DEM - with D4 it has to move only in 90 degree directions. However a main disadvantage of D8 is that the slope calculation for diagonals is different due to the longer distance from a central cell to a diagonal than a D4 neighbour.

Furthermore, D4 allows an elegant solution for parallel calculation of the scheme as fluxes of water need only be calculated (in both directions) for two neighbours N and W - and array values updated when calculated - rather than having to pass these values to a temporary array to be updated when all values across the domain have been calculated.

In actual modelling situations, the differences are moot. In all the examples shown here channels are represented by 1 or more cells wide meaning the principal advantage of diagonal movement is made redundant. We have not added the above discussion to the paper - as we think it possibly detracts or muddles the principal message - but it is explained in this response to reviewers which will remain on record in the interactive discussion.

*Results*

*For the final layout of the paper, can the maps and graphs for each case study be placed closer to the text of the case study? The UK results are fine, but the NZ and Brazil results are placed further and further from their respective sections. It would be better for the reader if the supporting maps and graphs were closer to the text.*

Changed

*The inundation maps (Fig 5, P13; Fig 7, P16; Fig 9, P19) show a very good overview of where the water sources for the inundation are coming from and how they are mixing. Would it possible to have a scale or legend item showing their respective depths? The text specifies*

*that the darker colours represent deeper depths, but a darkness-depth scale/legend item for the individual colours would be useful.*

We have added a colour scale to indicate depth for unmixed (i.e. red, green and blue) and selected combinations where appropriate. We have altered the figure caption to explain this.

*P14 L220-224: This section outlines the implications of knowing where water that is likely to contain pollutants is being deposited, and its effects on environment and human health. This discussion can be further expanded as this is a very important issue for water resource management. The abstract could also be updated to include one sentence or so about how the algorithm contributes to the mitigation/assessment of water quality issues.*

We have added a sentence to the end of the abstract to highlight this.

> This method enables water tracing 10 with a minimal computational overhead, allowing users of the LISFLOOD-FP method to address environmental issues relating to water sources and mixing, such as water quality and contamination problems.

Further, we have expanded the discussion (L330-337).

> Furthermore, the ability to combine both point source (e.g. from combined sewer outflows) and diffuse source (e.g. fields) contaminant sources within a computationally fast model opens up many opportunities to simulate water quality and pathogen/contaminant issues. Assessments of water sources may be especially useful for other fluvial or estuarine sites with similar human health considerations to the examples presented here. For example, Robins et al. (2019) assessed water quality risks from viral dispersal from the Conwy catchment, northWales, 335 and highlighted the importance of river flow contributions to exposure risk. Therefore, this simple and efficient water tracing algorithm within the open source CAESAR-Lisflood model provides a powerful tool for the studying water quality and contaminant issues on environmental and human health.

*Similar to the New Zealand application, does the Brazil application also have similar water quality issues? Do the Solimões and Purus rivers have similar or differing water quality and how would it affect downstream processes? Have there been water quality issues associated with flooding in the New Zealand and Brazil case studies?*

The issues are somewhat different, as the proportion of urban runoff is insignificant. However, there is an issue of human health in the basin which is mercury contamination from gold mining activities. We have added a sentence to note this and the possible information the method could provide for this - although we also note that sediment tracing is likely needed (L268-271):

> Further, it may enable a rapid assessment of issues of importance for human health, such as identifying areas which may be susceptible to mercury contamination from gold mining activities in the Amazon basin, for which the dynamics of fluvial and pluvial waters are key factors (Maurice-Bourgoin et al., 2003; Maia et al., 2009). However, for a robust analysis, it is likely that additional tracing of sediment erosion, transportation and deposition is required (e.g. Haddadchi et al., 2019).

*Although the processing time for the case studies is not comparable because of their differing timescales, it would be nice to have a summary table/overview of the three case studies taking about modelling domain size, grid size and number of cells, timescale, time taken to run the simulation, etc.*

Table 2 has been added to summarise each of these, and also provides a computational cost normalised by the simulation period and number of flooded cells, which makes the computation time more comparable.

*Discussion*

*As mentioned previously, it would be good to see discussion about the advantages of considering four flow directions, and if there would be significant changes if the D8 or Dinfinity flow directions are incorporated into the model.*

We have commented on this question earlier in the response to reviewers - but would like to add at this point that here we are enhancing an existing - long standing (since 2013 and earlier) hydraulic routing method (Lisflood-FP) that operates only on D4. Therefore discussion of D8 or Dinfinity routing methods is not in our view strictly relevant here.

*It would be good to have more discussions about the implications for water quality/contaminant issues on the environment health and human health, and how the algorithm can contribute to the mitigation of water quality issues. It helps underscore the contribution of the algorithm to modelling and to water resource management.*

We have added a section to the discussion highlighting this application. Thank you for the suggestion (L330-337).

*Testing the model*

*Please see the supplementary PDF for my notes about testing the model. I encountered some problems with running the Planar Test Case and the supplementary PDF shows the screenshots I encountered.*

*Please also note the supplement to this comment:*

*https://gmd.copernicus.org/preprints/gmd-2021-340/gmd-2021-340-RC1-supplement.pdf*

We have double checked with the supplemental material - and following the instructions have been able to run the planar test case and the Carlisle example.

However - the description of the supp material and how to use it was not as clear as it could have been. Therefore, we have changed/updated our supplemental material by both changing the description and incorporating the tracing components in the latest version of

the CAESAR-Lisflood model. This version contains some functional updates (not operational - not affecting model outcomes) and allows readers/users to have access to the most up to date version of the code. We have also placed the revised supp material in an updatable Zenodo archive ( https://zenodo.org/record/7589023) so any future changes can be clearly flagged up.

**Reviewer 2:**

*The authors have presented a water source tracing approach for hydrodynamic models used in flood inundation studies. The demonstrated method in this study is also independent of the hydraulic formulation and therefore has the potential to be used in other hydrodynamic/hydraulic models. The paper is well-written, and the formulation of the proposed methodology is presented neatly. The three case studies were demonstrated with complete details and strengthened the quality of the presented work. I feel the paper can be accepted with minor revisions after addressing the concerns presented below.*

*The wiki section of caesar-lisflood says "In the file tab - there are no additional boxes, but the tracer boxes have been removed. Tracer was rarely used yet added quite some complexity to the code, so for now has been removed." It looks like perhaps some earlier version of the model has already some sort of tracing mechanism with the caesar-lisflood model. If yes, how the current mechanism is different from the earlier one and why is it not even mentioned once in the manuscript?*

*https://sourceforge.net/p/caesar-lisflood/wiki/Moving%20From%20CAESAR%20to%20CAESARlisflood/*

Here the reviewer has looked at the main CAESAR-Lisflood repository rather than the Zenodo site flagged in the paper that contains the updated water tracing code. As in our response to reviewer 1 above - we have modified and clarified the supplemental material containing the code and examples.

To answer the reviewers question, an earlier version of CAESAR had a **sediment** tracing component for looking at the movement of contaminated sediment (see https://doi.org/10.1130/0091-7613(2003)031<0451:MLCIRS>2.0.CO;2 ) that is fundamentally different application - and science/coding issue.

We have not addressed this directly in the text - as again we don't want to confuse the main message of the paper (and the reviewer found outside of the paper supplemental material). However, if the editors wish this to be addressed we can add a sentence outlining/clarifying previous tracing functions.

*Why tracking algorithm is not implemented in the pure LISFLOOD-FP model and implemented in the caesar-lisflood model, when the focus of the study was only on "LISFLOOD-FP model of flood inundation".*

We have implemented this tracing method in CL rather than Lisflood-FP as CL is fully open source and integrates a GUI as well as a hydrological and morphodynamic model. However, the way the paper is structured with our pseudocode examples this is a straightforward task for people to add to Lisflood-FP (and other variants using similar methods). There are now several versions/evolutions of Lisflood-FP and so our examples presented here and the method are demonstrated within CL, but of course applicable to other similar models.

We have added a section at the end of the introduction to make this point clear (L44-49):

> We have used CAESAR-Lisflood for this purpose as the software is fully open source integrating a GUI as well as the hydraulic methods from LISFLOOD-FP. This enables our visualisation methods to be incorporated, but does not limit our tracing method to CAESAR-Lisflood. The equations and pseudo-code examples provided make it a straightforward task for users and researchers to add this functionality to LISFLOOD-FP (and other variants using similar methods).

*In the abstract, the line "A key advantage of the formulation developed is that the number of water sources which may be traced is limited only by computational considerations." is too general (especially in the field of hydrodynamics) and does not look appropriate as a main novelty of the proposed methodology.*

We have modified to say "The number of water sources that may be traced is limited only by computational considerations" (L4)

*The introduction section needs a lot of improvement. What was done and their brief motivation is only presented in the current manuscript. The relevant studies (especially make one section for related water source tracing studies in computational models) and different choices made in this study should be thoroughly discussed. Although differences in the approach exist between the current work and the work related to water source tracking presented in Qi et al. (2021, 2022), I would like to see authors to highlight/compare the advantages of the current (online) approach against the integrated multimodel (offline) water source tracking presented in Qi et al. (2021, 2022).*

Thank you for highlighting these additional references. We have added a short description of Qi et al's approach to the introductory section, which involves using pollutant tracers assigned to sources within PCSWMM model, then routed downstream using its pollutant transport scheme. An additional module developed then accounts for the relative mass of each tracer in each downstream catchment. Our approach is different in that we do not use tracers, and add the source tracking directly to a computationally efficient model code, avoiding the need for complex model codes. This has been highlighted (L30-36).

Qi, W., Ma, C., Xu, H., & Zhao, K. (2022). Urban flood response analysis for designed rainstorms with different characteristics based on a tracer-aided modeling simulation. Journal of Cleaner Production, 355, 131797.

Qi, W., Ma, C., Xu, H., Chen, Z., Zhao, K., & Han, H. (2021). Low impact development measures spatial arrangement for urban flood mitigation: an exploratory optimal framework based on source tracking. Water Resources Management, 35(11), 3755-3770.

———

**Astrid Kerkweg (executive editor, GMD) comment:**

*Dear authors,*

*in my role as Executive editor of GMD, I would like to bring to your attention our Editorial version 1.2: https://www.geosci-model-dev.net/12/2215/2019/*

*This highlights some requirements of papers published in GMD, which is also available on the GMD website in the 'Manuscript Types' section: http://www.geoscientific-model-development.net/submission/manuscript_types.html*

*In particular, please note that for your paper, the following requirement has not been met in the Discussions paper:*

- *"The main paper must give the model name and version number (or other unique identifier) in the title."*

*If I understand correctly the CAESAR-Lisflood model (version v1.8f) includes the Lisflood-FP model mentioned in the title. Would it be possible to provide a version number for LISFLOOD-FP in the title of your manuscript (in the revised submission to GMD).*

*Yours,*

*Astrid Kerkweg*

We have modified the title to include the software version for CAESAR-Lisflood as follows:

"Tracing and visualisation of contributing water sources in the LISFLOOD-FP model of flood inundation (within CAESAR-Lisflood version 1.9j-WS)"

This corresponds with the version in the updated Zenodo repository. With respect to LISFLOOD-FP, we do not refer to a software version number as it is the methodology within that model that is of interest. At the start of section 2.1 (L51), we note that CAESAR-Lisflood implements the inertial formulation of LISFLOOD-FP - cited as Bates et al. 2010. However, we also note that a key advantage is that the method is independent of the hydraulic formulation (abstract and L318), meaning that it could be implemented while using the other hydraulic methods within LISFLOOD-FP (there are several formulations within that model code).

---

## Author Response (AR2)

Dear Editor,

Thank you for the opportunity to address the one issue remaining from the reviewers' comments. As described below, previously we had not included a description to the earlier version as this was focussed on sediment tracing, not water source tracing, and we did not want to confuse the main message of our paper. However, we suggested that we could add some text if needed and this we have now done, as requested. We are very pleased that both reviewers have assessed the paper as having either good or excellent scientific significance and quality.

Below we provide the discussion background and a detailed response.

[Reviewer comments in black/ italic; responses in blue]

**Referee #1 (Report #2)**

Accepted as is. Excellent scientific significance and quality; good scientific reproducibility and presentation quality. No additional comments.

We are very grateful to the reviewer for their assessment of our paper.

**Referee #2 (Report #1)**

Accepted subject to minor revisions. Good scientific significance, quality and reproducibility, and good presentation quality.

Comment:

*Thank you for effort making the changes as per the different suggestions you had received. However, during my review, I raised a concern regarding the novelty of the proposed CL model with tracing mechanism and inquired about the distinctiveness and innovations incorporated in the current work compared to existing ones. I found your response ("that is fundamentally different application - and science/coding issue") somewhat equivocal and not entirely satisfactory. Apart from this concern, I would like to acknowledge that all other queries have been adequately addressed, and I am satisfied with the overall quality of the work.*

We are very grateful to the reviewer for their assessment of our paper and glad to hear that, with one exception, we have addressed all issues raised in the previous review. We thank the reviewer for highlighting their concern regarding a previous version of the model (CAESAR) which incorporated a form of sediment tracing that is not within the current version (CAESAR-Lisflood). We have sought to clarify the differences and have included a brief description, as below. The relevant comments from the original review and the response are included here for context.

Review comment (from 16 Oct 2022):

> *The wiki section of caesar-lisflood says "In the file tab - there are no additional boxes, but the tracer boxes have been removed. Tracer was rarely used yet added quite some complexity to the code, so for now has been removed." It looks like perhaps some earlier version of the model has already some sort of tracing mechanism with the caesar-lisflood model. If yes, how the current mechanism is different from the earlier one and why is it not even mentioned once in the manuscript?*
>
> *https://sourceforge.net/p/caesar-lisflood/wiki/Moving%20From%20CAESAR%20to%20CAESARlisflood/*

Response (from 9 Feb 2023; underline added to show quoted text):

> Here the reviewer has looked at the main CAESAR-Lisflood repository rather than the Zenodo site flagged in the paper that contains the updated water tracing code. As in our response to reviewer 1 above - we have modified and clarified the supplemental material containing the code and examples.
>
> To answer the reviewers question, an earlier version of CAESAR had a sediment tracing component for looking at the movement of contaminated sediment (see https://doi.org/10.1130/0091-7613(2003)031<0451:MLCIRS>2.0.CO;2 ) that is fundamentally different application - and science/coding issue.
>
> We have not addressed this directly in the text - as again we don't want to confuse the main message of the paper (and the reviewer found outside of the paper supplemental material). However, if the editors wish this to be addressed we can add a sentence outlining/clarifying previous tracing functions.

The review comment refers to the CAESAR-Lisflood webpage which provides user advice for moving from the earlier CAESAR model (focussed on sediment transport, steady state flow assumed, designed for simulation long term landscape evolution) to the current CAESAR-Lisflood (combined sediment transport and unsteady flow model). The key differences in the approaches are described in detail by Coulthard et al. (2013). The focus in the current paper is on tracing of water sources using the hydrodynamic model, independently of sediment routing. This would not have been possible with the CAESAR model due to its steady state flow formulation, as Coulthard et al. 2013 acknowledge. However, for completeness, we acknowledge that a reference to the earlier approach for sediment tracing would be useful.

On L44 of the manuscript, we have included a new paragraph to describe the previous work from Coulthard and Macklin (2003), and highlighted the difference with the approach here:

> In earlier work with CAESAR, Coulthard and Macklin (2003) demonstrated how sediment eroded from mining waste deposits could be traced downstream. The model worked by including different types of sediments which were used to represent contaminated and uncontaminated sediments as separate arrays for each sediment diameter included; during erosion and deposition of different sediment sizes, equal proportions of contaminated and uncontaminated sediment were transported. This enabled the prediction of patterns and levels of floodplain contamination, over a period of ~400 years. However, the approach was limited to sediment tracing only and did not account for different sources of water, which may carry contaminants with it. As noted by Coulthard et al. (2013), the inclusion of an unsteady flow formula within CAESAR-Lisflood has enabled the possibility, for example, to simulate water balances and solute fluxes using the model code. Here, we present the formulation of a simple methodology to enable this functionality by accounting for the source of water within model cells throughout the simulation.
>
> We use only the hydraulic and hydrological functionality of CAESAR-Lisflood to demonstrate the tracing method and visualisation of water sources, independently of sediment routing. We have used CAESAR-Lisflood for this purpose… etc.

We hope that this sufficiently clarifies the differences between the two approaches.

References:

Coulthard, T. J., & Macklin, M. G. (2003). Modeling long-term contamination in river systems from historical metal mining. *Geology*, *31*(5), 451-454.

Coulthard, T. J., Neal, J. C., Bates, P. D., Ramirez, J., de Almeida, G. A., & Hancock, G. R. (2013). Integrating the LISFLOOD-FP 2D hydrodynamic model with the CAESAR model: implications for modelling landscape evolution. *Earth Surface Processes and Landforms*, *38*(15), 1897-1906.

---

## Author Response (AR3)

Several minor corrections have been made to the manuscript as follows:

Line 2: "Water source tracing can provide additional insight into flood dynamics by accounting for flow pathways of each model boundary condition".

Line 18: "Dottori et al., 2022" [updated reference]

Lines 70, 84, 137, 373, 379: "Eqn." changed to "Eq."

Lines 70, 84, 137, 373, 379: use of "Eqn." changed to "Eq."

Line 84: "Following this, once flows between cells are calculated and depths updated, our water tracing method proceeds as follows:"

Line 103: "…using the notation of Eq. (1) and

Line 149: "This is used as part of the solution to Eq. (6) and (7) to update…"

Line 177: "However, using Eq. (15)…"

Line 201: changed "0.57% to 0.5%" to "0.5% to 0.57%"

Line 208: "The simulation began on 08 January 2005, 00:00 , for 120 hours"

Line 209: "Flood water mixing is shown in RGB colour space, using Eq. (15) to…"

Line 222: " intertidal area"

Line 224: "…New Brighton/ Southshore). It has been designated…"

Line 234: "…close to the mean annual flood level (Environment Canterbury, 2023), and Heathcote River peaked at 14:00 with 28.1 m3/s of flow, an Annual Exceedance Probability of less than 0.1 (Environment Canterbury, 2023)." [updated references]

Line 239: "…rivers 1.87 and 1.04 m3/s, respectively (LAWA, 2023)." [added year]

Line 259: "…recognised as a key factor…"

Line 270: "…through yellow (equal mix) …"

Line 286: "as per Eq. (14)"

Line 338: "… as each cell is treated as fully mixed …"

Line 342: "   contaminants  within a …"

Line 381: "   according to Eq. (1)"